# Echoes of Dormancy: Anomic Aphasia Unveils Neurocysticercosis Reactivation in a Patient on Semaglutide

**DOI:** 10.3390/neurosci6020040

**Published:** 2025-05-05

**Authors:** Marcos Osorio Borjas, Robert J. Hernandez, Angelo Lopez-Lacayo, Dalina Laffita Perez, Yanie Oliva, Julio Mercado, Hussain Hussain

**Affiliations:** 1Department of Internal Medicine, HCA Florida Kendall Hospital, Miami, FL 33175, USA; marcos.osorioborjas@hcahealthcare.com (M.O.B.); angelo.lopezlacayo@hcahealthcare.com (A.L.-L.); dlaffita2014@health.fau.edu (D.L.P.); yanie.oliva@hcahealthcare.com (Y.O.); julio.mercado@hcahealthcare.com (J.M.); 2Department of Infection Disease, HCA Florida Kendall Hospital, Miami, FL 33175, USA; roberthernandezmd@gmail.com; 3Department of Internal Medicine, Charles E. Schmidt College of Medicine, Florida Atlantic University, Boca Raton, FL 33431, USA

**Keywords:** *Taenia solium*, neurocysticercosis, albendazole, brain, semaglutide

## Abstract

Neurocysticercosis (NCC), a parasitic infection caused by *Taenia solium* larvae, remains a leading cause of acquired epilepsy worldwide, particularly in regions with inadequate sanitation and healthcare access. We present a case of NCC reactivation in a 64-year-old female who developed anomic aphasia—a rare manifestation of NCC—decades after her initial diagnosis. The patient’s clinical course was complicated by a potential trigger of semaglutide, which potentially attenuated the protective inflammatory response maintained by astrocytes and microglia, leading to the reactivation of dormant cysts. Brain imaging confirmed localized cystic changes, and treatment with antiparasitic agents and corticosteroids led to marked clinical improvement. This case highlights the complexity of NCC reactivation, highlighting the interplay of metabolic, immune, and parasitic factors. It emphasizes the need for vigilance in managing patients with dormant infections and investigating potential risks associated with novel therapeutic agents like GLP-1 receptor agonists. Further research is essential to unravel the mechanisms linking metabolic modulation to parasitic reactivation, offering insights into prevention and treatment strategies.

## 1. Introduction

Neurocysticercosis (NCC) is the most common parasitic infection of the central nervous system (CNS) and a leading cause of acquired epilepsy worldwide, particularly in regions with inadequate sanitation and poor access to healthcare [1,2]. It is caused by the larval stage (cysticercus) of *Taenia solium* (TS), a cestode that primarily affects humans and pigs [3,4]. NCC is endemic in regions with poor sanitation and close human–pig interactions, particularly in Latin America, sub-Saharan Africa, and parts of Asia [2]. The disease poses a significant burden on public health due to its association with epilepsy, neurological disability, and socioeconomic consequences.

In the United States, the incidence of NCC ranges from approximately 1.5 to 5.8 cases per 100,000 individuals, with the highest prevalence observed among Hispanic immigrants, particularly those from endemic regions such as Mexico, Central America, and South America [1,4,5,6]. The majority of cases in the U.S. occur in individuals who have either immigrated from or traveled to endemic areas where TS transmission remains a public health concern. Due to improved sanitation and stricter pork industry regulations, autochthonous transmission within the U.S. is rare; however, person-to-person transmission through fecal–oral contamination can still occur in households with asymptomatic tapeworm carriers [6]. Urban centers with large immigrant populations, such as Los Angeles, Houston, and New York City, report a disproportionately higher number of NCC cases [7]. Globally, NCC is a major cause of epilepsy and neurological morbidity, with an estimated 2.5 to 8.3 million people affected annually. The burden of disease is highest in endemic regions of Central and South America, sub-Saharan Africa, and parts of Asia, particularly India, Nepal, China, and Southeast Asia [6,8]. In these areas, factors such as poor sanitation, open defecation, lack of access to clean water, and free-ranging pig farming contribute to the continued transmission of TS. The World Health Organization (WHO) recognizes NCC as a neglected tropical disease (NTD), emphasizing the need for improved surveillance, preventive measures, and access to diagnostic and treatment modalities, particularly in low-resource settings where healthcare infrastructure is limited [9].

The life cycle of TS involves two hosts: pigs as the intermediate host and humans as the definitive host. Humans acquire adult tapeworm infection (taeniasis) by consuming undercooked pork containing cysticerci. Autoinfection or ingestion of eggs excreted in feces leads to larval migration to the CNS, resulting in NCC [4]. The ingested eggs hatch in the intestine, penetrate the intestinal wall, and disseminate via the bloodstream to various organs, including the brain, where they form cystic lesions [4,5,6].

NCC presents with diverse neurological symptoms depending on the location and severity of cystic involvement. Seizures are the most common manifestation, arising from localized inflammation or direct mechanical irritation of the brain that changes the neuronal impulses [10]. Persistent or reactivated infections may result in progressive cognitive decline, resembling dementia syndromes [10]. Focal neurological deficits, such as hemiparesis and vision loss, occur based on the cyst’s position [10]. Ventricular involvement can lead to hydrocephalus, obstructing cerebrospinal fluid flow and causing elevated intracranial pressure [4]. Additionally, neuropsychiatric symptoms, including depression, anxiety, and psychosis, are observed, particularly in chronic cases [11,12,13]. Anomic aphasia is a rare manifestation of NCC reactivation, occurring when cysts affect Wernicke’s area [1,10,13,14]. The occurrence of anomic aphasia in such cases raises intriguing questions about its underlying mechanisms. It suggests potential disruptions in neural networks or neurotransmitter signaling pathways, possibly involving secondary effects of inflammation or altered connectivity between remote brain regions.

The pathophysiology of NCC is highly dependent on the stage of cysticerci development and the host’s immune response [10]. In the acute phase, newly formed cysts initially evade immune detection by secreting anti-inflammatory molecules. However, as they degenerate, they elicit a strong immune response characterized by inflammation, perilesional edema, and gliosis, often manifesting clinically as seizure activity [2]. In the chronic stage, dormant cysts may reactivate years after the initial infection, triggering a delayed immune response mediated by cytokines and chemokines, which contribute to neuronal damage [6]. Persistent neuroinflammation and gliosis around calcified cysts have been linked to recurrent seizures, chronic epilepsy, and neurodegeneration [8,15]. Additionally, prolonged inflammatory activity has been associated with the accumulation of pathological proteins such as beta-amyloid and tau, which disrupt neuronal function and may contribute to cognitive decline, dementia, and psychological disorders [16]. In cases where cysticerci localize to the subarachnoid space or ventricular system, the inflammatory response can lead to complications such as arachnoiditis, hydrocephalus, and increased intracranial pressure, often requiring surgical intervention [16].

Persistent neuroinflammation may play a role in preventing the reactivation of NCC [15,16]. Consequently, any factors that influence or suppress the inflammatory response could potentially trigger the activation of dormant cysticerci. Semaglutide, a glucagon-like peptide-1 (GLP-1) receptor agonist commonly prescribed for type 2 diabetes and obesity, has been associated with various adverse effects, ranging from mild to severe [17]. Gastrointestinal symptoms, including nausea, vomiting, diarrhea, constipation, and abdominal pain, are the most frequently reported side effects, though these symptoms are generally transient and improve with continued use [17]. More serious but less common adverse effects include pancreatitis, gallbladder disease, and an increased risk of diabetic retinopathy complications, particularly in individuals with a history of poorly controlled diabetes [18]. Additionally, animal studies have suggested a potential association between semaglutide and thyroid C-cell tumors, though the clinical relevance of this finding in humans remains uncertain [18]. Other reported concerns include an increased heart rate and a possible decline in renal function in susceptible individuals [19]. Given the relatively recent introduction of semaglutide, ongoing post-marketing surveillance is essential to identify any emerging adverse effects. Therefore, we present a case of NCC activation in a patient currently taking semaglutide.

## 2. Case Presentation

A 64-year-old Hispanic female presented with persistent anomic aphasia for several weeks, accompanied by intermittent occipital headaches and paresthesia in the left upper and lower extremities. She denied any motor deficits or other systemic symptoms. On examination, her vital signs were within normal limits except for elevated blood pressure (181/85 mmHg). Her past medical history was significant for NCC and obesity, diagnosed in 1996 after experiencing seizures following travel to the Dominican Republic, where she had consumed undercooked meat. Brain imaging at that time confirmed NCC, and she was treated with a single course of albendazole in 1998. Her seizures were well controlled with phenytoin and carbamazepine, later transitioning to levetiracetam for improved management. She had also undergone bariatric surgery in 2016 and had been receiving semaglutide for weight reduction therapy over the past six months. The initial laboratory workup, including inflammatory markers (Table 1), was largely unremarkable except for a positive enzyme-linked immunosorbent assay (ELISA) for TS. Brain imaging performed in 2024 revealed NCC reactivation, characterized by localized inflammation and cystic changes following semaglutide therapy [Figure 1]. Notably, prior brain imaging from 2021, before semaglutide initiation, had shown no evidence of active NCC [Figure 2].

The patient was started on praziquantel 100 mg/kg/day for two weeks, albendazole 400 mg twice daily for one month, and dexamethasone 8 mg per day for two weeks, while continuing levetiracetam 1 g twice daily. Within five days of treatment initiation, her symptoms improved significantly, with resolution of anomic aphasia. She was subsequently discharged with follow-up care arranged through neurology and infectious disease services. Given the potential role of semaglutide in triggering NCC reactivation, discontinuation of the medication was also considered.

This case highlights the complexity of NCC reactivation and emphasizes the need for vigilant monitoring of latent infections in patients undergoing new medical therapies or significant physiological changes. Furthermore, it raises important considerations regarding the immunomodulatory effects of GLP-1 receptor agonists and their potential implications for chronic infections.

## 3. Discussion

The reactivation of NCC involves complicated mechanisms driven by parasitic degeneration and host immune responses [20]. Initially, the larvae remain dormant within protective barriers that prevent significant immune detection, allowing the infection to persist asymptomatically for years [1,8,10,15]. These barriers shield the parasite from the host’s immune system, maintaining a balance that avoids severe inflammatory reactions. Reactivation often occurs due to factors that disrupt immune equilibrium, such as immunosuppression from HIV/AIDS, corticosteroid use, organ transplantation, or the physiological changes of pregnancy [1,2,8,10,11,15,16]. Aging or coexisting infections can also diminish immune surveillance, allowing dormant cysticerci to degenerate [11,15]. This degeneration breaks the protective barriers, releasing parasitic antigens into surrounding neural tissue.

The release of these antigens prompts a robust immune response, leading to significant local inflammation. Immune cells release pro-inflammatory cytokines, including TNF-α, IL-6, and IL-1β, which amplify the inflammatory cascade [10,21,22]. Microglia and astrocytes become activated, contributing to the breakdown of the blood–brain barrier [10,21]. This exacerbates inflammation as immune cells infiltrate the central nervous system, further damaging neural tissue. The chronic inflammatory environment caused by reactivation can lead to structural damage in the brain, such as the formation of granulomas, calcifications, and gliosis [21,22]. These changes may disrupt neuronal networks, leading to seizures, focal neurological deficits, or cognitive decline [21]. Persistent inflammation also affects cerebral vasculature, further compounding neurological symptoms.

Semaglutide could play a significant role in the reactivation of NCC through a different proposed mechanism, further investigation is warranted. Astrocytes and microglia play an important role in the quiescence of the parasite within the brain through surrounding inflammation and eventually calcification [21,22]. Semaglutide may attenuate such inflammation, which could result in the weakening of the protective wall, ultimately increasing angiogenesis and the activation of TS [23].

Semaglutide, a GLP-1 receptor agonist, may contribute to the reactivation of neurocysticercosis (NCC), but there is limited direct evidence linking GLP-1 agonists to this effect, warranting further research. However, the brain’s innate immune cells, astrocytes and microglia, are crucial in maintaining the dormancy of TS cysticerci [22]. These cells create a protective inflammatory environment that encapsulates the parasites, often leading to calcification and chronic dormancy [23].

Semaglutide has demonstrated anti-inflammatory properties, particularly in metabolic and neurodegenerative conditions [24]. This attenuation of inflammation, while beneficial in reducing chronic inflammatory damage, might inadvertently compromise the host’s ability to maintain the integrity of the cyst’s encapsulation [25]. Reduced inflammation could weaken the protective immune barrier surrounding the cysts, disrupting quiescence [24,25]. Additionally, GLP-1’s role in promoting angiogenesis might further alter the microenvironment, increasing vascularization around the cyst and potentially activating dormant parasites [23].

Recent studies have indicated that semaglutide affects the brain and may reduce inflammation, highlighting its potential therapeutic benefits for conditions such as Parkinson’s disease and Alzheimer’s disease [24,25,26]. Semaglutide has been reported as a potential cause of ischemic optic neuropathy [27].

The hypothesized interaction between semaglutide and NCC reactivation emphasizes the complex interplay between metabolic regulation, immune modulation, and parasitic persistence in the CNS. Future research should aim to expose these mechanisms and determine whether GLP-1 receptor agonists contribute to an increased risk of NCC reactivation, especially in patients with known dormant infections or a history of exposure to TS.

The treatment of reactivated NCC involves a combination of antiparasitic, anti-inflammatory, and symptomatic therapies. Albendazole (400 mg twice daily) and praziquantel (50–100 mg/kg/day) are the primary antiparasitic agents used to target viable cysticerci, typically administered for 10 to 30 days depending on disease severity and cyst burden [8]. However, the rapid destruction of cysts can provoke an intense inflammatory response, necessitating the concurrent use of corticosteroids such as dexamethasone (6–8 mg/day) or prednisone to mitigate perilesional edema and prevent neurological complications, including seizures and hydrocephalus [8]. Long-term antiepileptic therapy, often with levetiracetam or valproic acid, is required in patients with seizure activity [2]. In cases of extensive cyst burden, subarachnoid involvement, or ventricular obstruction, surgical interventions such as ventriculoperitoneal shunting or cyst excision may be necessary.

## 4. Conclusions

Reactivation of NCC represents a significant clinical challenge, marked by the resurgence of symptoms such as seizures, cognitive dysfunction, and focal neurological deficits, often after prolonged dormancy. This phenomenon occurs when dormant TS cysts within the central nervous system are reactivated due to immune system alterations, such as reduced inflammatory surveillance or increased vascularization around the cysts. Potential triggers include immunosuppressive states, metabolic changes, or inflammatory modulations, as seen with certain medications or systemic conditions. Understanding the mechanisms behind NCC reactivation is crucial for early diagnosis and intervention. Prompt identification and treatment with antiparasitic agents like albendazole or praziquantel, combined with supportive therapies, can mitigate neuroinflammation, reduce neurological complications, and improve patient outcomes. Prevention of NCC requires a multifaceted approach involving sanitation, hygiene, and public health interventions. Proper cooking of pork, improved sanitation to prevent fecal–oral transmission, and health education on food safety are essential strategies. Additionally, mass deworming programs targeting taeniasis in endemic regions have shown promise in reducing transmission. Efforts to control TS in pigs, such as vaccination and anthelmintic treatment, have also been explored as preventive measures.

## Figures and Tables

**Figure 1 neurosci-06-00040-f001:**
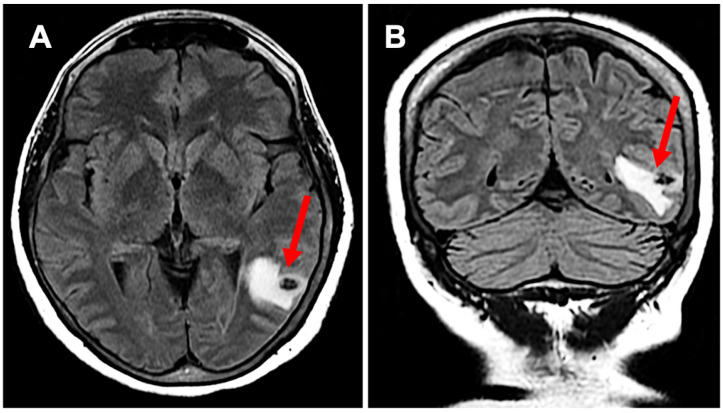
Subpanel (**A**) highlights neurocysticercosis surrounded by edema in a transverse view (red arrow), while subpanel (**B**) demonstrates the expansion of edema in a coronal view (red arrow); this activation developed while the patient was on semaglutide.

**Figure 2 neurosci-06-00040-f002:**
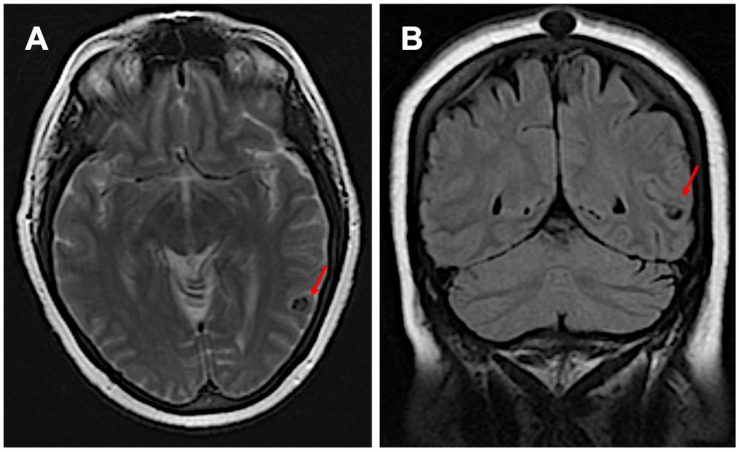
Imaging before starting semaglutide: subpanel (**A**) shows inactive neurocysticercosis without surrounding edema in the transverse view, while subpanel (**B**) presents the coronal view (red arrows).

**Table 1 neurosci-06-00040-t001:** Complete blood counts and comprehensive metabolic panel.

Sodium	137 (135–145)
Potassium	3.8 (3.5–5.3)
Chloride	105 (96–106)
Carbon dioxide	27 (35–45)
BUN	8 (<20)
Creatinine	0.63 (<1)
Glomerular filtration rate (GFR)	>90
Glucose	85
Lactic acid	0
Calcium	8.9 (8.5–10.2)
Magnesium	1.8 (1.7–2.2)
Total bilirubin	0.5 (<1.2)
Aspartate aminotransferase (AST)	26 (<35)
Alanine aminotransferase (ALT)	17 (<40)
Alkaline phosphatase	84 (44–147)
Triglycerides	62 (<150)
Cholesterol	169 (<200)
Low density lipoprotein (LDL)	81 (<100)
High density lipoprotein (HDL)	76 (>45)
White blood cell (WBC)	6.0 (4–10)
Red blood cell (RBC)	4.21 (4.3–5.6)
Hemoglobin	12.4 (11.6–15)
Hematocrit	37.1 (36–44)
Platelets	228 (150–450)
Prothrombin time (PT)	12.8 (11–13.5)
Partial thromboplastin time (PTT)	31 (25–35)
International normalization rate (INR)	1.1 (0.8–1.1)
Erythrocytes sedimentation rate	35 (<20)
C-reactive protein	3 (0.3–1)

## Data Availability

A request to the corresponding author is required to obtain any data.

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
