# Peer review of "Echoes of Dormancy: Anomic Aphasia Unveils Neurocysticercosis Reactivation in a Patient on Semaglutide"

_neurosci, 2025, doi:10.3390/neurosci6020040_

Round 1
Reviewer 1 Report
Comments and Suggestions for Authors
Overall: Interesting case report on potential effect of semaglutide on neurocysticercosis. Authors present MRI data and a discussion of potential effects of semaglutide. However, as there are limited data, the premises are mostly hypothetical. Additional patient data is needed and should be reported. Without those additional data, the report is incomplete.
Specific comments:
‘anomia” can have different meanings. Possibly ‘anomic aphasia’ would be more specific ?
Figure1 and 2: expand on the explanation /labeling of the figure in the legend, specific label the figure, e.g. A,B, C,D and specifically indicate which conditions these are pre treatment / control and disease. Current information provided is not sufficient.
line 89 do authors mean "may have lead to reactivation"
Cell and metabolic profiles of the patient are missing, e.g. including glucose values and should be included.
Are there any other data on this patient, e.g. inflammatory profile, neurofilament fragments, c-reactive protein in plasma etc. This should be included.
Additional information of the secondary effects of semaglutide should be mentioned in the introduction and followed up on in discussion.
Semaglutide not only reduces inflammation, e.g. affect the inflammasome /NLRP3 but also affects immune cell infiltration/transmigration into the brain. As we know now that the brain is NOT immune privileged, presence of immune cells, including plasma cells / antibody production, is needed to keep certain infections under control. Reduced presence of the right type of immune cells in the brain may lead to reactivation and/or expansion of infectious agents. These aspects are not clearly discussed.
Line 114, 118 , 120, 126 etc. Usually, new sentences start with start with capital letter, Thus, it these cases it should be S (not semaglutide). Not sure why authors do not choose this and use a lowercase letter.
Line 126: Semaglutide has (not have) or state ‘semaglutide and analogues have”.
Line 117: reference for the calcification needed
Author Response
We very much appreciate the opportunity to resubmit our manuscript, “Echoes of Dormancy: Anomia Unveils Neurocysticercosis Reactivation in a Patient on Semaglutide.” First, we thank the Reviewers for their positive and encouraging comments on our m/s and the many valuable suggestions for improvement. We have seriously considered all these concerns, and appropriate changes have been made in the revised m/s.
In this resubmission, we have responded to all the issues raised by the reviewers and made appropriate changes. We have submitted modified version changes have been identified in red color and strikeouts.
Reviewer 1
Overall: Interesting case report on potential effect of semaglutide on neurocysticercosis. Authors present MRI data and a discussion of potential effects of semaglutide. However, as there are limited data, the premises are mostly hypothetical. Additional patient data is needed and should be reported. Without those additional data, the report is incomplete.
Specific comments:
‘anomia” can have different meanings. Possibly ‘anomic aphasia’ would be more specific ?
- Thank you very much for the valuable suggestion. Now we changed to “anomic aphasia as per the suggestion.
Figure1 and 2: expand on the explanation /labeling of the figure in the legend, specific label the figure, e.g. A,B, C,D and specifically indicate which conditions these are pre treatment / control and disease. Current information provided is not sufficient.
- As per the instruction, we changed both legends. We much appreciated the comment
line 89 do authors mean "may have lead to reactivation"
- Yes, it was an error, now we changed. Thank you
Cell and metabolic profiles of the patient are missing, e.g. including glucose values and should be included.
- Thank you for the suggestion, we included a table with all the values
Are there any other data on this patient, e.g. inflammatory profile, neurofilament fragments, c-reactive protein in plasma etc. This should be included.
- Thank you for the suggestion, we included a table with all the values
Additional information of the secondary effects of semaglutide should be mentioned in the introduction and followed up on in discussion.
- Now we have added 2 paragraphs in the introduction and discussion as per your suggestion. Thank you
Semaglutide not only reduces inflammation, e.g. affect the inflammasome /NLRP3 but also affects immune cell infiltration/transmigration into the brain. As we know now that the brain is NOT immune privileged, presence of immune cells, including plasma cells / antibody production, is needed to keep certain infections under control. Reduced presence of the right type of immune cells in the brain may lead to reactivation and/or expansion of infectious agents. These aspects are not clearly discussed.
- Kindly note that semaglutide reduces inflammation, which is sometimes essential for maintaining certain infections in an inactive state. For example, in pulmonary tuberculosis, inflammation is necessary to keep the granuloma stable and prevent bacterial reactivation. A reduction in inflammation may lead to the reactivation of bacteria or parasites. We hope this answer your question, the text is well written by a native English speaker and has reviewed one more time to make sure the content is preserved.
Line 114, 118 , 120, 126 etc. Usually, new sentences start with start with capital letter, Thus, it these cases it should be S (not semaglutide). Not sure why authors do not choose this and use a lowercase letter.
- Yes, it was an error, now we changed. Thank you
Line 126: Semaglutide has (not have) or state ‘semaglutide and analogues have”.
- Yes, it was an error, now we changed. Thank you
Line 117: reference for the calcification needed
- We now added the suitable reference as per the suggestion. Thank you so much.
Reviewer 2 Report
Comments and Suggestions for Authors
This manuscript aims to describe a case of NCC reactivation after Semaglutide administration. This is a well-written easy-to-follow manuscript.
I have a few minor issues to address before publication.
line 79. Point out the period between images 1 and 2. How long apart the were images taken?
lines 85-89. Add duration and treatment doses. Also, add how long it took for the patient to return to normal.
lines 120-122. this sentence is confusing: "semaglutide, a GLP-1 receptor agonist, may play a significant role in the reactivation of NCC through several proposed mechanisms, though direct evidence linking GLP-1 agonists to NCC reactivation remains scarce and warrants further investigation." Please, clarify
I found several times "semaglutide" starting a sentence instead of Semaglutide. Please, correct it
I just have some questions for the authors (do not need to add them to the manuscript)
What do you think about the original TS cyst treatment given to the patient? Could this have been the cause of the persistence of this larva over the years? Do you think that the treatment that was recently given could have effectively eliminated the parasite?
Author Response
We very much appreciate the opportunity to resubmit our manuscript, “Echoes of Dormancy: Anomia Unveils Neurocysticercosis Reactivation in a Patient on Semaglutide.” First, we thank the Reviewers for their positive and encouraging comments on our m/s and the many valuable suggestions for improvement. We have seriously considered all these concerns, and appropriate changes have been made in the revised m/s.
In this resubmission, we have responded to all the issues raised by the reviewers and made appropriate changes. We have submitted modified version changes have been identified in red color and strikeouts.
This manuscript aims to describe a case of NCC reactivation after Semaglutide administration. This is a well-written easy-to-follow manuscript.
I have a few minor issues to address before publication.
line 79. Point out the period between images 1 and 2. How long apart the were images taken?
- Thank you for the comment, in 2021 no activation noted. While in 2024 there is activation noted. We added the information in the text.
lines 85-89. Add duration and treatment doses. Also, add how long it took for the patient to return to normal.
- Kindly, we now added the information as per the suggestion.
lines 120-122. this sentence is confusing: "semaglutide, a GLP-1 receptor agonist, may play a significant role in the reactivation of NCC through several proposed mechanisms, though direct evidence linking GLP-1 agonists to NCC reactivation remains scarce and warrants further investigation." Please, clarify
- Thank you, no we have modified the sentence.
I found several times "semaglutide" starting a sentence instead of Semaglutide. Please, correct it
- Yes, it was an error, now we changed. Thank you
I just have some questions for the authors (do not need to add them to the manuscript)
What do you think about the original TS cyst treatment given to the patient? Could this have been the cause of the persistence of this larva over the years? Do you think that the treatment that was recently given could have effectively eliminated the parasite?
- Thank you. The standard of care for this condition is albendazole, and the patient was treated according to the guidelines. However, it can sometimes be challenging to fully eliminate the parasite, as certain serotypes have the ability to form cysts in the brain. In some cases, reactivation can occur in the future due to immune disturbances. A similar concept applies to TB reactivation in the lungs when TNF-alpha inhibitors are used, as they reduce inflammation and allow the bacteria to become more active. I hope this explanation clarifies the concept.